# Gaussian Mixture Vector Quantization with Aggregated Categorical Posterior

## Abstract

The vector quantization is a widely used method to map continuous representation to discrete space and has important application in tokenization for generative mode, bottlenecking information and many other tasks in machine learning. Vector Quantized Variational Autoencoder (VQ-VAE) is a type of variational autoencoder using discrete embedding as latent. We generalize the technique further, enriching the probabilistic framework with a Gaussian mixture as the underlying generative model. This framework leverages a codebook of latent means and adaptive variances to capture complex data distributions. This principled framework avoids various heuristics and strong assumptions that are needed with the VQ-VAE to address training instability and to improve codebook utilization. This approach integrates the benefits of both discrete and continuous representations within a variational Bayesian framework. Furthermore, by introducing the *Aggregated Categorical Posterior Evidence Lower Bound* (ALBO), we offer a principled alternative optimization objective that aligns variational distributions with the generative model. Our experiments demonstrate that GM-VQ improves codebook utilization and reduces information loss without relying on handcrafted heuristics.

## 1 Introduction

Variational autoencoders (VAEs) (Kingma & Welling, 2013) were originally designed for modeling continuous representations; however, applying them to discrete latent variable models is challenging due to non-differentiability. A common solution is to use gradient estimators tailored for discrete latent variables. The REINFORCE estimator (Williams, 1992) is an early example, providing an unbiased estimate of the gradient but suffering from high variance. Alternatively, methods such as the Gumbel-Softmax reparameterization trick (Jang et al., 2017; Maddison et al., 2017) allow for a continuous relaxation of categorical distributions. While these methods introduce bias into the gradient estimation, they offer the benefit of significantly lower variance, thereby improving training stability.

Vector Quantized Variational Autoencoders (VQ-VAEs) (Van Den Oord et al., 2017) extend the VAE framework to discrete latent spaces by discretizing continuous representations through a codebook via straight-through estimator (STE) (Bengio et al., 2013). Beyond the inherent variance-bias tradeoff in gradient estimation, VQ-VAEs are known to suffer from codebook collapse, wherein all encodings converge to a limited set of embedding vectors, resulting in the underutilization of many vectors in the codebook. This phenomenon diminishes the information capacity of the bottleneck. Takida et al. (2022); Williams et al. (2020) hypothesized that deterministic quantization is the cause of codebook collapse and introduced stochastic sampling, leading to a entropy term in the log-likelihood lower bound. While high entropy is generally beneficial, it is inherently incompatible with Gumbel-Softmax gradient estimation. Various handcrafted heuristics have been proposed to mitigate this issue, including batch data-dependent k-means (Łańcucki et al., 2020), replacement policies (Zeghidour et al., 2021; Dhariwal et al., 2020), affine parameterization with alternate optimization (Huh et al., 2023), and entropy penalties (Yu et al., 2023). However, as these heuristics do not derive from the evidence lower Bound (ELBO), they cannot be unified within the variational Bayesian framework, rendering them ad-hoc solutions lacking a coherent foundation in variational inference.

Figure 1: **Overview of GM-VQ**. First, the encoder deterministically maps the input to proxy latents, which are then used to retrieve corresponding codewords from the codebook and generate noise. The codewords and noise are then combined to form the continuous latents. Finally, these continuous latents are passed through the decoder to produce the final output.

In our work, we propose a Gaussian mixture prior based on VQ-VAE within a variational Bayesian framework, namely *Gaussian Mixture Vector Quantization* (GM-VQ, see Figure 1), combining the benefits of both discrete and continuous representations while avoiding handcrafted heuristics and strong assumptions. Additionally, to optimize the model and ensure compatibility with the gradient estimation errors inherent to Gumbel-Softmax, we modify the ELBO by replacing the conditional categorical posterior with an aggregated categorical posterior, resulting in an novel lower bound, the *Aggregated Categorical Posterior Evidence Lower Bound* (ALBO), which minimizes estimation error while preserving codebook utilization. Concretely, our contributions are as follows:

- To the best of our knowledge, we are the first to apply the Gaussian mixture prior formulation on VQ-VAE with strict adherence to the variational Bayesian framework.

- We introduce *Aggregated Categorical Posterior Evidence Lower Bound* (ALBO), which is explicitly designed to be compatible with Gumbel-Softmax gradient estimation.

- We conduct experiments demonstrating improved codebook utilization and reduced information loss without relying on handcrafted heuristics.

## 2 PRELIMINARIES

### 2.1 DETERMINISTIC VQ-VAE

The VQ-VAE (Van Den Oord et al., 2017), based on the VAE framework (Kingma & Welling, 2013), learns discrete latent representations via vector quantization (Gray, 1984). For simplicity, we represent discrete latents with a single random variable $\hat{\mathbf{z}}$ here, though in practice, we extract latent features of various dimensions.

Given an encoder output $\hat{\mathbf{z}}$ from a high-dimensional input $\mathbf{x}$ and a latent embedding space (codebook) $\mathbf{M} \in \mathbb{R}^{C \times L}$, composed of $C$ row vectors $\boldsymbol{\mu}_i \in \mathbb{R}^L$, selects the discrete latent variable (codeword) as $j = \arg\min_i \|\hat{\mathbf{z}} - \boldsymbol{\mu}_i\|$, yielding the approximate posterior distribution $q(\mathbf{c} \mid \mathbf{x}) \in \mathbb{R}^C$:

$$q(\mathbf{c} \mid \mathbf{x}) = [\mathbf{1}_{\mathrm{i=j}}]_{i=1}^C = \begin{cases} 1, & \text{if } i = \arg\min_i \|\hat{\mathbf{z}} - \boldsymbol{\mu}_i\|, \\ 0, & \text{otherwise.} \end{cases} \tag{1}$$

Based on the one-hot distribution $q(\mathbf{c} \mid \mathbf{x})$, we have query vector $\mathbf{c}_q = q(\mathbf{c} \mid \mathbf{x})$ for codebook, the quantized latent representation is expressed as $\mathbf{z}_c = \mathbf{c}_q^T \mathbf{M} = \boldsymbol{\mu}_j \in \mathbb{R}^L$.

To handle the non-differentiability of the quantization process, the straight-through estimator (STE) (Bengio et al., 2013) is applied, with the assumption $\frac{\partial \mathbf{z}_c}{\partial \hat{\mathbf{z}}} = \boldsymbol{I}$ to allow gradient flow. This assumption holds reasonably well when $\mathbf{z}_c$ does not deviate significantly from $\hat{\mathbf{z}}$; however, greater devia-

tions introduce increased bias. To mitigate this issue and enhance gradient estimation, an additional discretization loss was introduced:

$$\mathcal{L}_{\text{discretization}}(\hat{\mathbf{z}}, \mathbf{z}_c) = \|\hat{\mathbf{z}} - \text{sg}[\mathbf{z}_c]\|^2 + \alpha \cdot \|\text{sg}[\hat{\mathbf{z}}] - \mathbf{z}_c\|^2. \qquad (2)$$

Here, sg[·] represents the stop-gradient operator, and $\alpha$ adjusts the balance between minimizing the discrepancies between $\hat{\mathbf{z}}$ and $\mathbf{z}_q$.

## 2.2 STOCHASTIC VQ-VAE

One problem of VQ-VAE is that the learned discrete representation uses only a fraction of the full capacity of the codebook, a problem known as codebook collapse. To solve this problem, stochastic sampling was introduced by modifying the approximate posterior , shifting from a one-hot representation to a distribution proportional to the negative squared distance between the encoder output $\hat{\mathbf{z}}$ and the codewords $\boldsymbol{\mu}_c$(Roy et al. (2018); Sønderby et al. (2017); Shu & Nakayama (2017)):

$$q(\mathbf{c} \mid \mathbf{x}) = \text{Softmax}\left(-\frac{\|\hat{\mathbf{z}} - \boldsymbol{\mu}_c\|^2}{2\sigma^2}\right). \qquad (3)$$

To estimate gradients, the Gumbel-Softmax trick (Jang et al., 2017; Maddison et al., 2017) approximates the categorical distribution as $q^g(\mathbf{c} \mid \mathbf{x}) = \text{Softmax}_\tau(\log q_i(\mathbf{c} \mid \mathbf{x}) + \mathbf{g}_i)$, where $\mathbf{g}_i$ are independent and identically distributed (i.i.d.) samples from Gumbel(0,1). The index $j$ is selected as $j = \arg\max_i q_i^g(\mathbf{c} \mid \mathbf{x})$, resulting in the quantized representation $\mathbf{c}_q = [\mathbf{1}_{i=j}]_{i=1}^C$.

In this framework, the gradient assumption shifts from $\frac{\partial \mathbf{z}_c}{\partial \hat{\mathbf{z}}} = \boldsymbol{I}$ to $\frac{\partial \mathbf{c}_q}{\partial q^g(\mathbf{c}|\mathbf{x})} = \boldsymbol{I}$. This gradient estimation requires a low entropy in $q(\mathbf{c} \mid \mathbf{x})$ to be accurate . When entropy $q(\mathbf{c} \mid \mathbf{x})$ has a higher entropy the output distribution becomes more uniform. In this scenario, the Gumbel-Softmax trick outputs values that are less peaked, leading to noisy estimates for gradient-based optimization because the model becomes less certain about which category is the most likely. This uncertainty increases the inaccuracy in the gradient estimates. Thus, with a fixed temperature, high entropy in $q(\mathbf{c} \mid \mathbf{x})$ leads to gradient estimation errors. However, this leads to a problem.

Assuming a uniform prior, the variational bound is:

$$-\log p(\mathbf{x}) \le \mathbb{E}_{\mathbf{c} \sim q(\mathbf{c}|\mathbf{x})}\left[-\log p(\mathbf{x} \mid \mathbf{c})\right] - H(q(\mathbf{c} \mid \mathbf{x})) + \log C, \qquad (4)$$

which introduces a negative entropy loss that promotes a high-entropy posterior distribution, conflicting with low entropy requirement in Gumbel-Softmax trick and resulting in incompatible high gradient estimation bias.

## 3 GM-VQ: GAUSSIAN MIXTURE VECTOR QUANTIZATION

To solve the problem mentioned above, we propose a stochastic vector quantization method using using Gaussian mixture distribution, which we refer to as *Gaussian Mixture Vector Quantization* (GM-VQ), with the following encoding and decoding process:

$$\mathbf{x} \xrightarrow{E_{\boldsymbol{\theta}}} \hat{\mathbf{z}} \xrightarrow{\mathbf{M}} \mathbf{c} \xrightarrow{\boldsymbol{\varepsilon}} \mathbf{z} \xrightarrow{D_{\boldsymbol{\phi}}} \tilde{\mathbf{x}} \qquad (5)$$

In our framework, GM-VQ comprises an encoder $E_{\boldsymbol{\theta}}$ and a decoder $D_{\boldsymbol{\phi}}$, parameterized by the deep neural network weights $\boldsymbol{\theta}$ and $\boldsymbol{\phi}$, respectively, and interconnected via a codebook $\mathbf{M}$, which contains the means of Gaussian mixtures. The term $\boldsymbol{\varepsilon}$ denotes the Gaussian noise introduced into the process.

Unlike previous works adapted VQ-VAE with Gaussian mixture priors (Takida et al., 2022; Williams et al., 2020), which directly transmit categorical representations to the decoder, our method explicitly adds noise to the categorical variables during training, feeding continuous latent variables into the decoder. While Liu et al. (2021) theoretically demonstrate that discrete representations can enhance generalization and robustness during evaluation, the codewords in our codebook are continuously updated throughout training, leading to a non-static categorical representation. By explicitly introducing noise, our method allows the decoder to adapt in parallel with the evolving codebook.

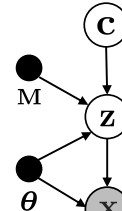 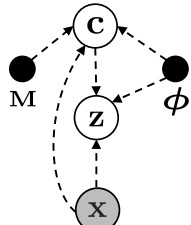

Figure 2: Probabilistic Graphical Model depicting the Gaussian Mixture Vector Quantization (GM-VQ) for the generative model (left) and the inference model (right). The codebook M plays a dual role, being shared between both the generative and inference models.

### 3.1 GENERATIVE MODEL

The generative model is defined by the joint distribution:

$$p(\mathbf{x}, \mathbf{z}, c) = p(\mathbf{x} \mid \mathbf{z})\, p(\mathbf{z} \mid c)\, p(c), \tag{6}$$

where $\mathbf{x} \in \mathbb{R}^D$ represents the observed data, $\mathbf{z} \in \mathbb{R}^L$ is a continuous latent variable, and $c \in \{1, 2, \ldots, C\}$ is a discrete latent variable indicating the mixture component. The latent variable $\mathbf{z}$ follows a Gaussian mixture distribution, with the means stored in a codebook $\mathbf{M} \in \mathbb{R}^{C \times L}$, where each row $\boldsymbol{\mu}_c$ represents a codeword for component $c$.

First, the discrete variable $c$ is sampled from a categorical distribution, where $\boldsymbol{\pi}$ represents the prior probabilities of the mixture components, typically assumed to be uniformly distributed. Then, given $c$, the continuous latent variable $\mathbf{z}$ is drawn from a multivariate Gaussian distribution with mean $\boldsymbol{\mu}_c$ and isotropic covariance matrix $\sigma_{\mathbf{z}}^2 \boldsymbol{I}$. As $\sigma_{\mathbf{z}}^2$ approaches zero, this setup converges to deterministic quantization, similar to VQ-VAE.

Finally, the observed data $\mathbf{x}$ is generated conditionally on $\mathbf{z}$, where the decoder $D_{\boldsymbol{\theta}}$ maps the continuous latents $\mathbf{z}$ to the mean of the observation distribution, modeled as a Gaussian with fixed variance $\sigma_{\mathbf{x}}^2 \boldsymbol{I}$.

### 3.2 VARIATIONAL INFERENCE AND POSTERIOR ESTIMATION

To approximate the intractable posterior $p(c, \mathbf{z} \mid \mathbf{x})$, we employ a variational posterior $q(c, \mathbf{z} \mid \mathbf{x})$ following standard variational autoencoder methods. Without simplifying assumptions, we employ variational inference with an approximate posterior. We use the chain rule to factorize the posterior:

$$q(c, \mathbf{z} \mid \mathbf{x}) = q(c \mid \mathbf{x})q(\mathbf{z} \mid \mathbf{x}, c), \tag{7}$$

allowing for conditional dependencies between $\mathbf{z}$ and $c$ given $\mathbf{x}$. To achieve this, we introduce the following variational distributions:

1. **Variational Distribution over** $c$:

$$q(c \mid \mathbf{x}) = \text{Categorical}\big(\boldsymbol{\pi}(\mathbf{x})\big), \tag{8}$$

   where $\boldsymbol{\pi}(\mathbf{x}) = (\boldsymbol{\pi}_1(\mathbf{x}), \boldsymbol{\pi}_2(\mathbf{x}), \ldots, \boldsymbol{\pi}_C(\mathbf{x}))$ are the posterior probabilities over the mixture components, parameterized by an encoder $E_{\boldsymbol{\theta}}$ and codebook $\boldsymbol{M}$.

2. **Variational Distribution over** $\mathbf{z}$:
   We model $q(\mathbf{z} \mid \mathbf{x}, c)$ as a multivariate Gaussian distribution centered at the codeword $\boldsymbol{\mu}_c$, with a covariance $\boldsymbol{\Sigma}_c(\mathbf{x})$ that depends on $\mathbf{x}$:

$$q(\mathbf{z} \mid \mathbf{x}, c) = \mathcal{N}\big(\mathbf{z}; \boldsymbol{\mu}_c, \boldsymbol{\Sigma}_c(\mathbf{x})\big), \tag{9}$$

   where $\boldsymbol{\Sigma}_c(\mathbf{x}) = \sigma_c^2(\mathbf{x})\boldsymbol{I}$ is an isotropic covariance matrix.

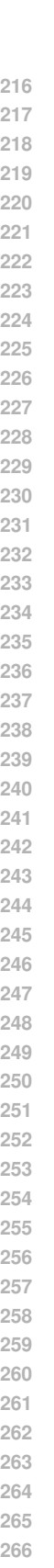
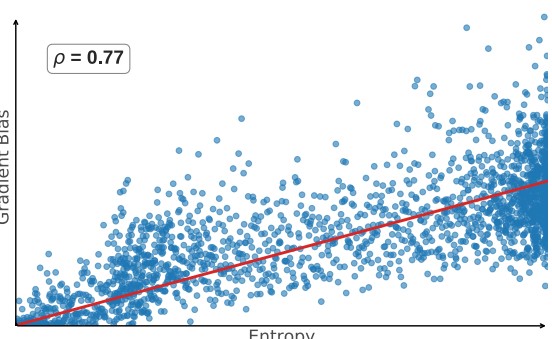

Given a non-linear network, unnormalized logits corresponding to varying entropy levels are fed into the model, and the bias between Gumbel-Softmax gradient estimation and the exact gradient was calculated. A strong Pearson correlation ($\rho = 0.77, p \leq 0.001^{***}$) shows that gradient estimation errors increase with rising entropy. For more implementation details, Appendix A.1.

Figure 3: Gradient Bias vs. Entropy Relationship

### 3.3 TRAINING OBJECTIVE AND OPTIMIZATION

To train the model, we typically maximize the ELBO using Gumbel-Softmax gradient estimation. However, the presence of entropy in ELBO leads to poor Gumbel-Softmax gradient estimation(see Figure 3). Given that entropy can be particularly detrimental to Gumbel-Softmax gradient estimation, it is critical to mitigate its impact during training.

To resolve this, Yu et al. (2023) introduced two contrasting entropy measures derived from mutual information (Krause et al., 2010):

$$\mathcal{L}_{\text{entropy}} = H(q(c \mid \mathbf{x})) - H(\mathbb{E}_{\mathbf{x} \sim p(\mathbf{x})} q(c \mid \mathbf{x})). \tag{10}$$

In the context of vector quantization, the first term of this loss reduces uncertainty when mapping observed data $\mathbf{x}$ to c and the second term encourages more discrete latent variables to be used, which decreases the codebook collapse problem.

However, this approach remains an additional heuristic that cannot be fully incorporated into the variational Bayesian framework. To address this limitation, we propose an alternative ELBO formulation.

#### 3.3.1 AGGREGATED CATEGORICAL POSTERIOR EVIDENCE LOWER BOUND

We introduce the *Aggregated Categorical Posterior Evidence Lower Bound* (ALBO) as an alternative to the traditional ELBO:

$$\mathcal{E}_{\text{ALBO}}(x) = \mathbb{E}_{q(c) \, q(\mathbf{z}|\mathbf{x})} \left[ \log \frac{p(\mathbf{x}, \mathbf{z}, c)}{q(c)} \right] \leq \log p(\mathbf{x}). \tag{11}$$

The ALBO provides a lower bound on the log-likelihood $\log p(\mathbf{x})$ (for derivation, see Appendix A.2).

Here, $q(c)$ represents the marginal distribution over c, induced by the data distribution $p(\mathbf{x})$ and the approximate posterior $q(c \mid \mathbf{x})$, defined as $q(c) = \int q(c \mid \mathbf{x}) p(\mathbf{x}) \, d\mathbf{x}$. However, since the true data distribution $p(\mathbf{x})$ is not accessible in practice, we have to rely on a finite dataset instead. To balance computational complexity and accuracy, we use a **mini-batch approximation**. For a mini-batch $\mathcal{B}$ of size $|\mathcal{B}|$, the marginal distribution $q(c)$ is approximated as:

$$q^{(\mathcal{B})}(c) = \frac{1}{|\mathcal{B}|} \sum_{\mathbf{x} \in \mathcal{B}} q(c \mid \mathbf{x}). \tag{12}$$

This mini-batch approximation is computationally efficient and well-suited for stochastic optimization in deep learning.

For simplicity, $\sigma_{\mathbf{x}}^2$ and $\sigma_{\mathbf{z}}^2$ are typically fixed and not treated as learnable parameters. Thus, the objective function is constructed by minimizing the negative ALBO, resulting in the GM-VQ loss $\mathcal{L}_{\text{GM-VQ}}(\mathbf{x})$:

$$\mathcal{L}_{\text{GM-VQ}}(\mathbf{x}) = \mathbb{E}_{q(\mathbf{z}|\mathbf{x})}\|\mathbf{x} - D_{\boldsymbol{\theta}}(\mathbf{z})\|^2 + \gamma \cdot \mathcal{L}_{\text{reg}}(\mathbf{x}), \tag{13}$$

where the regularization term $\mathcal{L}_{\text{reg}}(\mathbf{x})$ is defined as:

$$\mathcal{L}_{\text{reg}}(\mathbf{x}) = \mathbb{E}_{q(\text{c})q(\mathbf{z}|\mathbf{x})}\|\mathbf{z} - \boldsymbol{\mu}_c\|^2 + \beta \cdot D_{\text{KL}}(q(c)\|p(c)), \tag{14}$$

with $\beta$ and $\gamma$ as non-negative hyperparameters controlling the balance between reconstruction and regularization. Detailed derivation can be found in Appendix A.3.

This loss function consists of three main components:

- $\mathbb{E}_{q(\mathbf{z}|\mathbf{x})}\|\mathbf{x} - D_{\boldsymbol{\theta}}(\mathbf{z})\|^2$: Reconstruction loss, standard in autoencoder frameworks.
- $\mathbb{E}_{q(\text{c})q(\mathbf{z}|\mathbf{x})}\|\mathbf{z} - \boldsymbol{\mu}_c\|^2$: Latent regularization, ensures alignment between the latent variables and the learned codewords $\boldsymbol{\mu}_c$.
- $D_{\text{KL}}(q(\text{c})\|p(\text{c}))$: Entropy term, enhances overall codebook utilization and prevents individual $q(c \mid \mathbf{x})$ from drifting towards high entropy, thereby reducing gradient estimation errors.

### 3.3.2 PARAMETERIZATION OF THE VARIATIONAL DISTRIBUTIONS

**Variational Categorical Distribution** $q(\text{c} \mid \mathbf{x})$   To compute $q(\text{c} \mid \mathbf{x})$, we use the encoder $E_{\boldsymbol{\phi}}$ to generate a proxy representation $\hat{\mathbf{z}}(\mathbf{x}) \in \mathbb{R}^L$ and raw weights $\hat{\mathbf{r}}(\mathbf{x}) \in \mathbb{R}^L$, which are then activated into positive weights $\hat{\mathbf{w}}(\mathbf{x})$ using the Softplus function $\zeta(\cdot)$:

$$\hat{\mathbf{z}}(\mathbf{x}), \; \hat{\mathbf{r}}(\mathbf{x}) = E_{\boldsymbol{\phi}}(\mathbf{x}), \quad \hat{\mathbf{w}}(\mathbf{x}) = \zeta\big(\hat{\mathbf{r}}(\mathbf{x})\big). \tag{15}$$

The unnormalized log probabilities $\boldsymbol{l}_c(\mathbf{x})$ are then computed based on the Mahalanobis-like distance between $\hat{\mathbf{z}}(\mathbf{x})$ and each codeword $\boldsymbol{\mu}_c$:

$$\begin{aligned}
\boldsymbol{l}_c(\mathbf{x}) &= -\frac{1}{2} \left(\hat{\mathbf{z}}(\mathbf{x}) - \boldsymbol{\mu}_c\right)^\top \text{diag}(\hat{\mathbf{w}}(\mathbf{x})) \left(\hat{\mathbf{z}}(\mathbf{x}) - \boldsymbol{\mu}_c\right) \\
&= -\frac{1}{2} \sum_{i=1}^{L} \hat{\mathbf{w}}_i(\mathbf{x}) \left(\hat{\mathbf{z}}_i(\mathbf{x}) - \boldsymbol{\mu}_{c,i}\right)^2.
\end{aligned} \tag{16}$$

The posterior probabilities $\boldsymbol{\pi}_c(\mathbf{x})$ are obtained via softmax:

$$\boldsymbol{\pi}_c(\mathbf{x}) = \text{Softmax}\big(\boldsymbol{l}_c(\mathbf{x})\big) = \frac{\exp\big(\boldsymbol{l}_c(\mathbf{x})\big)}{\sum_{c'=1}^{C} \exp\big(\boldsymbol{l}_{c'}(x)\big)}. \tag{17}$$

This formulation ensures that components with codewords closer to $\hat{\mathbf{z}}(\mathbf{x})$ have higher posterior probabilities.

**Variational Continuous Distribution** $q(\mathbf{z} \mid \mathbf{x}, \text{c})$   The variance $\boldsymbol{\sigma}_c^2(\mathbf{x})$ in $q(\mathbf{z} \mid \mathbf{x}, \text{c})$ reflects the uncertainty in assigning $\mathbf{x}$ to component c. It is parameterized based on the squared distance between the encoder's output $\hat{\mathbf{z}}(\mathbf{x})$ and the codeword $\boldsymbol{\mu}_c$:

$$\boldsymbol{\sigma}_c^2(\mathbf{x}) = \frac{\|\hat{\mathbf{z}}(\mathbf{x}) - \boldsymbol{\mu}_c\|^2/L}{2\sigma^2}, \tag{18}$$

where $\sigma^2$ is the scalar variance from the generative model and $L$ is the latent dimensionality.

This parameterization allows $q(\mathbf{z} \mid \mathbf{x}, \mathbf{c})$ to adapt its variance based on the distance between $\hat{\mathbf{z}}(\mathbf{x})$ and $\boldsymbol{\mu}_c$, ensuring a flexible representation of uncertainty. As proxy latent $\hat{\mathbf{z}}(\mathbf{x})$ approaches $\boldsymbol{\mu}_c$, the variance $\boldsymbol{\sigma}_c^2(\mathbf{x})$ decreases, indicating higher confidence in the assignment to component $c$. Conversely, when the distance increases, the variance grows, signaling greater uncertainty.

**Reparameterization and Codebook Update** To enable backpropagation through the discrete sampling of c, we use the Gumbel-Softmax reparameterization trick. The discrete latent variable c is computed as $j = \arg\max \operatorname{Softmax}_\tau(\log q(\mathbf{c} \mid \mathbf{x}) + \mathbf{g})$, where $\mathbf{g}$ are i.i.d. Gumbel(0,1) samples, yielding the quantized representation $\mathbf{c}_q = [\mathbf{1}_{i=j}]_{i=1}^C$.

For the continuous latent variable $\mathbf{z}$, we apply the standard VAE reparameterization: $\mathbf{z} = \boldsymbol{\mu}_c + \boldsymbol{\sigma}_c(\mathbf{x}) \odot \boldsymbol{\epsilon}$, with $\boldsymbol{\epsilon} \sim \mathcal{N}(0, \boldsymbol{I})$. In the ALBO framework, we sample from $q(\mathbf{z} \mid \mathbf{x})$, combining the quantized codeword and noise as $\mathbf{z} = \mathbf{c}_q^T \mathbf{M} + \boldsymbol{\sigma}_c(\mathbf{x}) \odot \boldsymbol{\epsilon}$.

This formulation ensures that all codewords $\boldsymbol{\mu}_c$ are naturally updated during optimization, preventing the codebook collapse problem without the need for additional commitment loss functions or exponential moving averages. It enables the model to manage deviations from the codewords in a controlled and efficient manner.

## 4 RELATED WORK

Variational Autoencoders (VAEs) (Kingma & Welling, 2013) comprise a generative model and a recognition model, bridged by a latent variable typically modeled with a multivariate Gaussian prior. While the generative component is well-known, the recognition model effectively learns continuous representations from data (Zhang et al., 2022; Yang et al., 2021; Zhao et al., 2017; Higgins et al., 2017).

To address discrete representation learning, Vector Quantized VAE (VQ-VAE) (Razavi et al., 2019; Van Den Oord et al., 2017) employs vector quantization (Gray, 1984) to discretize latent embeddings under a uniform prior. Since the quantization process is non-differentiable, techniques like the straight-through estimator (Bengio et al., 2013) are used to approximate gradients, introducing potential bias. VQ-VAE also incorporates a discretization loss to mitigate these issues, but challenges such as codebook underutilization and information loss remain. Sønderby et al. (2017); Shu & Nakayama (2017) introduced stochastic sampling based on the negative distance and applied Gumbel-Softmax for gradient estimation. Later, Karpathy (2021); Esser et al. (2021) proposed an encoder that directly outputs the posterior, applying Gumbel-Softmax without conditioning on the codebook.

While many existing VAEs utilize Gaussian mixture priors (Liu et al., 2023; Bai et al., 2022; Falck et al., 2021; Guo et al., 2020; Jiang et al., 2016; Dilokthanakul et al., 2016; Nalisnick et al., 2016), our approach is distinct in its close connection with vector quantization. Specifically, we reuse the means from the codebook, whereas in other GMM-based models, the posterior means are typically learned transiently, conditioned on different components or networks. Although Takida et al. (2022); Williams et al. (2020) also explore Gaussian mixture priors in VQ-VAE, they modify the reconstruction loss and feed discrete latents directly into the decoder, deviating from strict adherence to the ELBO. Furthermore, prior works often rely on simplifying assumptions on variational posterior, such as mean-field (Liu et al., 2023; Falck et al., 2021; Figueroa, 2017; Jiang et al., 2016) or Markovian assumptions (Takida et al., 2022; Williams et al., 2020).

Previous works (Tomczak & Welling, 2018; Hoffman & Johnson, 2016; Makhzani et al., 2015) have attempted to modify the ELBO framework by averaging the objective over data distribution, but they retain the original variational conditional distribution, leaving the entropy term of the conditional posterior it intact. In contrast, our approach is specifically motivated by the gradient estimation error in categorical latents by using an aggregated categorical posterior instead of the conditional categorical posterior.

Beyond these theoretical advancements, vector quantization (VQ) and its related concepts have also seen extensive applications across various domains. Here, we highlight some recent works. Uni-MoT (Zhang et al., 2024) introduces a VQ-driven tokenizer that converts molecules into molecular token sequences, while VQSynergy (Wu et al., 2024) integrates VQ for drug synergy prediction. In image and video generation, MAGVIT-v2 (Yu et al., 2023) applies VQ-VAE to achieve high-fidelity

reconstructions, and VAR (Tian et al., 2024) leverages VQ to advance visual autoregressive learning. These studies demonstrate the pivotal role of VQ in both theoretical advancements and practical applications.

## 5 EXPERIMENTS

In this section, we provide a comprehensive analysis of our experiments using the proposed GM-VQ model, focusing on its performance in image reconstruction tasks across two benchmark datasets CIFAR10 and CelebA.

### 5.1 EXPERIMENTAL SETUP

We use Gumbel-Softmax for gradient estimation throughout our experiments, adopting an annealing schedule similar to Takida et al. (2022); Jang et al. (2017). The temperature starts at 2.0 and is gradually reduced to 0.1 during training.

Our architecture and hyperparameters closely follow the setup in Huh et al. (2023). For the CIFAR10 dataset (32x32 image size), we use a compact architecture consisting of convolutional layers followed by vector quantization, similar to standard autoencoders. For CelebA (resized to 128x128), a deeper network is employed to manage the higher resolution. Both architectures incorporate a VQ layer to quantize the latent representations into discrete codes.

Models are trained for 100 epochs on both datasets, using a batch size of 256. We employ AdamW (Loshchilov, 2017) as the optimizer, with a maximum learning rate of 1e-2 for CIFAR10 and 1e-4 for CelebA. The learning rate follows a cosine decay schedule with linear warmup as in Huh et al. (2023), with 10 epochs of warmup starting at a factor of 0.2, followed by 90 epochs of cosine decay. All models are initialized using K-means clustering for codebook initialization, following Esser et al. (2021). We use 1024 codes with a latent dimension of 64 across all experiments, applying the same weight decay to both the encoder-decoder and the codebook. During evaluation, no noise is added to the reconstructed $\mathbf{x}$ for maximum likelihood estimation, and latents $\mathbf{z}$ are directly sampled from the codebook $\mathbf{M}$ without extra noise.

We report Mean Squared Error (MSE) as the metric for reconstruction quality, which measures the mean pixel-wise difference between the original and reconstructed images. Additionally, we report perplexity, defined as $2^{H(q)}$, to evaluate the diversity of codebook usage. Higher perplexity indicates a more balanced use of the available codes. Notably, this perplexity is not based on the entropy of individual codes $q(\mathbf{c} \mid \mathbf{x})$ but on the average entropy across a batch of categorical distributions $q^{(\mathcal{B})}(\mathbf{c})$, with perplexity computed per batch and then averaged across all batches.

We compare our GM-VQ model against several baseline methods commonly used in vector quantization-based image reconstruction. The primary baseline is the standard VQ-VAE (Van Den Oord et al., 2017). Variants include VQ-VAE + $l_2$ (Yu et al., 2021), which stabilizes training through $l_2$ normalization, and VQ-VAE + replace (Zeghidour et al., 2021; Dhariwal et al., 2020), which replaces unused code vectors with random embeddings to avoid codebook collapse. SQ-VAE (Takida et al., 2022) introduces stochastic quantization for improved code diversity, while Gumbel-VQVAE (Karpathy, 2021; Esser et al., 2021) employs Gumbel-softmax for smoother gradient updates. Lastly, VQ-VAE + affine + OPT (Huh et al., 2023) addresses codebook covariate shift with affine parameterization and alternating training. Our primary model, GM-VQ, was tuned by fixing $\beta = 1$ and selecting the best $\gamma$, while the variant GM-VQ + Entropy was tuned with higher entropy regularization ($\beta > 1$) and the fixed $\gamma$, promoting more balanced codebook usage.

### 5.2 PERFORMANCE COMPARISON

We evaluate the performance of GM-VQ on the CIFAR10 and CelebA datasets, comparing it against several baseline methods. Table 1 presents the results, using MSE for reconstruction accuracy and perplexity metrics.

In the CIFAR10 dataset, GM-VQ achieves an MSE of 3.13, a significant improvement over the standard VQVAE (MSE 5.65) and variants like VQVAE + $l_2$ (MSE 3.21) and VQVAE + replace (MSE 4.07). In terms of codebook utilization, GM-VQ achieves a perplexity of 731.9, considerably

| Method | CIFAR10 | | CELEBA | |
|---|---|---|---|---|
| | MSE ($10^{-3}$) ↓ | Perplexity ↑ | MSE ($10^{-3}$) ↓ | Perplexity ↑ |
| VQVAE | 5.65 | 14.0 | 10.02 | 16.2 |
| VQVAE + $l_2$ | 3.21 | 57.0 | 6.49 | 188.7 |
| VQVAE + replace | 4.07 | 109.8 | 4.77 | 676.4 |
| VQVAE + $l_2$ + replace | 3.24 | 115.6 | 4.93 | 861.7 |
| VQVAE + Affine | 5.15 | 69.5 | 7.47 | 112.6 |
| VQVAE + OPT | 4.73 | 15.5 | 7.78 | 30.5 |
| VQVAE + Affine + OPT | 4.00 | 79.3 | 6.60 | 186.6 |
| SQVAE | 3.36 | 769.3 | 9.17 | 769.1 |
| Gumbel-VQVAE | 6.16 | 20.3 | 7.34 | 96.7 |
| GM-VQ | 3.13 | 731.9 | 1.38 | 338.6 |
| GM-VQ + Entropy | 3.11 | 878.7 | 0.97 | 831.0 |

Table 1: Comparison of methods on CIFAR10 and CELEBA datasets using MSE and Perplexity metrics.

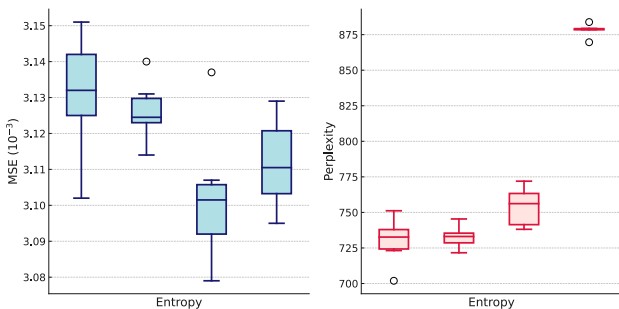

Figure 4: Box plots showing the impact of entropy regularization on reconstruction quality (MSE) and codebook utilization (Perplexity) for the GM-VQ model. The left panel demonstrates the general trend of decreasing MSE with increasing entropy, the right pane shows the rise in perplexity with higher perplexity.

higher than VQ-VAE + replace (perplexity 109.8), indicating more efficient and diverse code usage. This highlights GM-VQ's ability to mitigate codebook collapse and ensure robust code assignments.

On the CelebA dataset, GM-VQ excels with an MSE of 1.38, significantly outperforming the baseline VQVAE (MSE 10.02) and VQVAE + replace (MSE 4.77). GM-VQ also maintains strong codebook diversity with a perplexity score of 338.6. The GM-VQ + Entropy variant further enhances performance, achieving the lowest MSE of 0.97 and a perplexity of 831.0. This shows that entropy regularization effectively promotes balanced codebook usage without sacrificing reconstruction quality.

In summary, across both datasets, GM-VQ and GM-VQ + Entropy consistently outperform all baseline models in terms of both reconstruction accuracy and codebook utilization. These results demonstrate the model's robustness and its ability to maintain a balance between reconstruction fidelity and efficient code usage.

## 5.3 IMPACT OF ENTROPY REGULARIZATION

To further demonstrate the compatibility of the aggregated posterior with increased entropy, we conducted experiments across different entropy regularization levels using GM-VQ on the CIFAR10 dataset.

The box plots in Figure 4 provide a summary of how changes in entropy affect both Mean Squared Error (MSE) and Perplexity. In the left panel, we observe a general trend where MSE decreases

as entropy increases, indicating a tendency toward improved reconstruction quality, though not uniformly across all entropy levels. Meanwhile, the right panel shows that higher entropy promotes more effective codebook usage, with perplexity rising as entropy grows, reflecting more balanced code assignments.

This trend suggests that while increased entropy yields better codebook utilization (higher perplexity), it also drives improvements in reconstruction accuracy (lower MSE).

## 6 CONCLUSION

In summary, the GM-VQ framework extends the traditional VQ-VAE by incorporating a probabilistic structure grounded in a Gaussian mixture model. By employing the ALBO objective, we ensure optimization is well-suited to Gumbel-Softmax gradient estimation. Our empirical results demonstrate the effectiveness of GM-VQ, highlighting its ability to achieve a strong balance between reconstruction quality and codebook diversity.

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

# A APPENDIX

## A.1 IMPLEMENTATION DETAILS OF ENTROPY BIAS

To investigate the relationship between entropy and the bias in Gumbel-Softmax gradient estimation, we conducted targeted experiments.

We used a multi-layer perceptron (MLP) with hidden layers of size [50, 5], where the input consists of 10 possible actions and the output is a scalar. To obtain the exact gradients, we pass each categorical one-hot action through the non-linear decoder. For the Gumbel-Softmax gradient estimation, we repeated the experiment 50 times to compute the empirical average of the estimated gradients.

To assess the impact of entropy on bias, we varied the input entropy by applying softmax with different temperature ($\tau$) values to a fixed set of unnormalized logits, which determined the input probabilities for the MLP. This setup allowed us to analyze how changes in entropy influence the bias in Gumbel-Softmax gradient estimation.

## A.2 DERIVATION OF ALBO

The *Aggregated Categorical Posterior Evidence Lower Bound* (ALBO) provides a lower bound on the log-likelihood $\log p(\mathbf{x})$, similar in structure to the Evidence Lower Bound (ELBO) commonly used in variational inference. This bound is derived through the application of Jensen's inequality, as shown below:

$$
\begin{aligned}
\log p(\mathbf{x}) &= \log \mathbb{E}_{q(\mathrm{c})q(\mathbf{z}|\mathbf{x})} \left[ \frac{p(\mathbf{x}, \mathbf{z}, \mathrm{c})}{q(\mathrm{c})q(\mathbf{z} \mid \mathbf{x})} \right] \\
&\geq \mathbb{E}_{q(\mathrm{c})q(\mathbf{z}|\mathbf{x})} \left[ \log \frac{p(\mathbf{x}, \mathbf{z}, \mathrm{c})}{q(\mathrm{c})q(\mathbf{z} \mid \mathbf{x})} \right] \quad \text{(by Jensen's inequality)} \\
&= \mathbb{E}_{q(\mathrm{c})q(\mathbf{z}|\mathbf{x})} \left[ \log \frac{p(\mathbf{x}, \mathbf{z}, \mathrm{c})}{q(\mathrm{c})} \right] - \mathbb{E}_{q(\mathbf{z}|\mathbf{x})} \left[ \log q(\mathbf{z} \mid \mathbf{x}) \right] \\
&\geq \mathbb{E}_{q(\mathrm{c})q(\mathbf{z}|\mathbf{x})} \left[ \log \frac{p(\mathbf{x}, \mathbf{z}, \mathrm{c})}{q(\mathrm{c})} \right] \quad \text{(since } \mathbb{E}_{q(\mathbf{z}|\mathbf{x})} \left[ \log q(\mathbf{z} \mid \mathbf{x}) \right] \leq 0) \\
&= \mathcal{E}_{\mathrm{ALBO}}(\mathbf{x}).
\end{aligned}
\tag{19}
$$

By applying Jensen's inequality, the logarithm is moved inside the expectation, yielding a tractable lower bound. The term $\mathbb{E}_{q(\mathbf{z}|\mathbf{x})} \left[ \log q(\mathbf{z} \mid \mathbf{x}) \right]$ represents the entropy of the posterior over $\mathbf{z}$. Since

entropy is non-positive, it further tightens the bound, ensuring that $\mathcal{E}_{\text{ALBO}}(\mathbf{x})$ provides a meaningful approximation.

Thus, $\mathcal{E}_{\text{ALBO}}(\mathbf{x})$ serves as a valid lower bound on $\log p(\mathbf{x})$, similar to the ELBO in traditional variational inference. The key difference is in the aggregation of the categorical posterior over c, which offers better compatibility with probability-based gradient estimation.

## A.3 DERIVATION OF GM-VQ LOSS

Minimizing $\mathcal{L}_{\text{GM-VQ}}(\mathbf{x})$ ensures that the model effectively reconstructs the data while regularizing the latent distributions to align with the priors, thereby improving generalization. Below, we provide the detailed derivation of the GM-VQ loss.

We start by maximizing the $\mathcal{E}_{\text{ALBO}}(\mathbf{x})$:

$$
\begin{aligned}
&\arg\max_{\boldsymbol{\theta},\boldsymbol{\phi},\mathbf{M}} \mathcal{E}_{\text{ALBO}}(\mathbf{x}) \\
&= \arg\min_{\boldsymbol{\theta},\boldsymbol{\phi},\mathbf{M}} -\mathcal{E}_{\text{ALBO}}(\mathbf{x}) \\
&= \arg\min_{\boldsymbol{\theta},\boldsymbol{\phi},\mathbf{M}} -\mathbb{E}_{q(\mathbf{c})q(\mathbf{z}|\mathbf{x})} \left[ \log \frac{p(\mathbf{x},\mathbf{z},\mathbf{c})}{q(\mathbf{c})} \right] \\
&= \arg\min_{\boldsymbol{\theta},\boldsymbol{\phi},\mathbf{M}} \mathbb{E}_{q(\mathbf{c})q(\mathbf{z}|\mathbf{x})} \left[ -\log \frac{p(\mathbf{x}\mid\mathbf{z})\,p(\mathbf{z}\mid\mathbf{c})\,p(\mathbf{c})}{q(\mathbf{c})} \right] \\
&= \arg\min_{\boldsymbol{\theta},\boldsymbol{\phi},\mathbf{M}} \mathbb{E}_{q(\mathbf{z}|\mathbf{x})} \left[ -\log p(\mathbf{x}\mid\mathbf{z}) \right] + \mathbb{E}_{q(\mathbf{c})q(\mathbf{z}|\mathbf{x})} \left[ -\log p(\mathbf{z}\mid\mathbf{c}) \right] + D_{\text{KL}}(q(\mathbf{c})\|p(\mathbf{c})) \\
&= \arg\min_{\boldsymbol{\theta},\boldsymbol{\phi},\mathbf{M}} \mathbb{E}_{q(\mathbf{z}|\mathbf{x})} \left[ \frac{\|\mathbf{x}-D_{\boldsymbol{\theta}}(\mathbf{z})\|^2}{2\sigma_{\mathbf{x}}^2} \right] + \mathbb{E}_{q(\mathbf{c})q(\mathbf{z}|\mathbf{x})} \left[ \frac{\|\mathbf{z}-\boldsymbol{\mu}_c\|^2}{2\sigma_{\mathbf{z}}^2} \right] + D_{\text{KL}}(q(c)\|p(c)) \\
&= \arg\min_{\boldsymbol{\theta},\boldsymbol{\phi},\mathbf{M}} \mathbb{E}_{q(\mathbf{z}|\mathbf{x})}\|\mathbf{x}-D_{\boldsymbol{\theta}}(\mathbf{z})\|^2 + \frac{\sigma_{\mathbf{x}}^2}{\sigma_{\mathbf{z}}^2} \cdot \left( \mathbb{E}_{q(\mathbf{c})q(\mathbf{z}|\mathbf{x})}\|\mathbf{z}-\boldsymbol{\mu}_c\|^2 + 2\sigma_{\mathbf{z}}^2 \cdot D_{\text{KL}}(q(c)\|p(c)) \right)
\end{aligned}
\tag{20}
$$

Given variance terms $\sigma_{\mathbf{x}}^2$ and $\sigma_{\mathbf{z}}^2$ are fixed, we can replace them with positive hyperparameters $\beta$ and $\gamma$, respectively, to simplify this expression, where:

1. $\gamma = \frac{\sigma_{\mathbf{x}}^2}{\sigma_{\mathbf{z}}^2}$ controls the balance between reconstruction and latent regularization.

2. $\beta = 2\sigma_{\mathbf{z}}^2$ modulates the strength of the KL divergence regularization.

Thus, the GM-VQ loss can be rewritten as:

$$
\mathcal{L}_{\text{GM-VQ}}(\mathbf{x}) = \mathbb{E}_{q(\mathbf{z}|\mathbf{x})}\|\mathbf{x}-D_{\boldsymbol{\theta}}(\mathbf{z})\|^2 + \gamma \cdot \left( \mathbb{E}_{q(\mathbf{c})q(\mathbf{z}|\mathbf{x})}\|\mathbf{z}-\boldsymbol{\mu}_c\|^2 + \beta \cdot D_{\text{KL}}(q(c)\|p(c)) \right)
\tag{21}
$$

This formulation, with hyperparameters $\beta$ and $\gamma$, balances the reconstruction fidelity and the regularization of the latent space.

