# OpenReview forum: "Gaussian Mixture Vector Quantization with Aggregated  Categorical Posterior"
_ICLR.cc/2025/Conference — ICLR 2025 Conference Withdrawn Submission_

### Official Review · Reviewer_GRy6 · 2024-10-31

**Soundness:** 1
**Presentation:** 3
**Contribution:** 2
**Rating:** 3
**Confidence:** 5

**Summary:**

The paper proposes a new vector quantization method based on Gaussian Mixture Models (GMMs). Specifically, within the VAE framework, the authors first employ a GMM as the prior and then they design a specifically tailored variational inference arm to implement vector quantization, using a newly introduced ELBO variant called ALBO. Experiments are conducted on the CIFAR10 and CelebA datasets to demonstrate the effectiveness of the proposed method.

**Strengths:**

The paper is well written and easy to follow in general.

The proposed techniques are original to my knowledge.

**Weaknesses:**

The soundness of the proposed ALBO is questionable. See Questions below.

**Questions:**

The tightness of the ALBO in Eq. (11) is not discussed in detail. For example, ALBO is looser than ELBO. What are its advantages/disadvantages?

The parameterization of the variational inference arm should be elaborated. Intuitively, one would model the inference arm as $q(c|x)q(z|x,c)$ following Fig. 2; why use the aggregated $q(c)q(z|x)$ instead in Eq. (11)? What's the definition of $q(z|x)$?

The optimal inference arm in the ELBO framework is the posterior. What's the optimal solution for the inference arm in the ALBO framework?

What's the influence of the mini-batch approximation?

In Line 294, "$E_{\phi}$" is a typo.

---

### Official Review · Reviewer_FdUM · 2024-10-31

**Soundness:** 2
**Presentation:** 3
**Contribution:** 2
**Rating:** 5
**Confidence:** 3

**Summary:**

This work proposes a new formulation for describing VQ-VAE within a Bayesian framework using a Gaussian Mixture prior. This approach avoids the need for heuristics to ensure effective codebook utilization. The authors introduce an alternative training objective, the Aggregated Categorical Posterior Evidence Lower Bound (ALBO), designed to reduce Gumbel-Softmax gradient estimation errors while enhancing codebook usage.

**Strengths:**

- This work presents an alternative formulation for describing VQ-VAE within a Bayesian framework, avoiding heuristics.
- The method is clearly explained, and the authors provide context through a discussion of relevant state-of-the-art methods.
- The implementation and experimental details are thoroughly outlined.

**Weaknesses:**

- I believe the experimental evaluation of the method could be improved. While the authors compare their method with several state-of-the-art approaches, the evaluation focuses only on MSE for reconstruction and Perplexity for codebook usage. Including additional generation results would add value (does this model require two-stage training as the original VQ-VAE, or it is not necessary given the new formulation?). An ablation study on the effect of batch size, given its influence on the marginalization of q(c), could clarify its impact on performance (if there’s any). Further codebook analysis would also be beneficial; for instance, are the learned codebook vectors diverse, or are they quite similar, potentially explaining the higher entropy?
- The paper lacks a concluding discussion and does not explore potential limitations of the method or directions for future research, which I believe would add significant value.

**Questions:**

- The authors should review the notation throughout the paper, as there are typos that may difficult readability. For instance, the parameters of the Decoder neural network are inconsistently denoted as $\theta$ and $\phi$.
- In the graphical model of the generative model, why does $z$ depend on the parameters of the decoder? In its definition, $p(z|c)$ is not parameterized by the decoder.
- In section 3, the authors state ‘By explicitly introducing noise, our method allows the decoder to adapt in parallel with the evolving codebook.’ Could the authors elaborate on this idea? Does the noise act as additional regularization, helping to prevent the decoder from overfitting to a specific codebook?
- How does the performance of the method vary with respect to $\sigma^{2}_{z}$? As it approaches zero, the method converges to the deterministic quantization. I assume that there is a tradeoff in the amount of noise introduced in the method.
- How does the batch size affect the performance of the method, given its influence on the marginalization of $q(c)$?
- I think it would be beneficial to evaluate the entropy of $q(c∣x)$ (either during training or evaluation) to demonstrate that the proposed objective effectively prevents individual $q(c∣x)$ distributions from drifting toward high entropy, thereby reducing gradient estimation errors.
- How is the generation performance of this method? Does it need a two-stage training as the original VQ-VAE?
- What are the potential limitations of this method?
- I believe it would be beneficial to include an ablation study on the hyperparameters controlling the balance of the loss to characterize the training dynamics.

---

### Official Review · Reviewer_AGaB · 2024-11-01

**Soundness:** 2
**Presentation:** 2
**Contribution:** 2
**Rating:** 5
**Confidence:** 4

**Summary:**

This paper proposes a variant of VQ-VAE that utilizes stochastic vector quantization based on a Gaussian mixture prior.
The model is trained by optimizing a proposed variational lower bound of log-likelihood, referred to as ALBO, which is derived from a new graphical model within a variational Bayesian framework.
The proposed loss function is designed to be compatible with Gumbel-Softmax gradient estimation.
Experiments on two benchmark datasets demonstrate that the proposed method improves reconstruction accuracy and codebook utilization compared to baseline methods.

**Strengths:**

* The proposed algorithm is clearly explained.
* The variant of VQ-VAE is theoretically derived by a graphical model and a variational Bayesian framework.
* In the experiments, the proposed method is compared to a sufficient number of VQ-VAE baselines.

**Weaknesses:**

* The paper does not clearly demonstrate why the proposed method resolves the existing problems. Some parts could be explained more clearly.
  - In the Introduction, the paper claims, "we are the first to apply the Gaussian mixture prior formulation on VQ-VAE with strict adherence to the variational Bayesian framework." However, I find it challenging to identify the key differences compared to the work of Takida et al. (2022) and Williams et al. (2020) in this regard. In Section 4, the paper states, "they modify the reconstruction loss and feed discrete latents directly into the decoder, deviating from strict adherence to the ELBO." However, these operations are theoretically and naturally derived from their own proposed variational lower bound loss within the Bayesian framework. --> (Q1)
  - The main content does not clearly explain the key differences from previous methods or why the proposed framework resolves the issue of gradient estimation error. After reviewing the supplementary material in Section A.2, I noticed that the key difference lies in the aggregation of the categorical posterior over c, which does not conflict with the low entropy requirement for q(c|x) in the Gumbel-Softmax trick. (I think the statement in Section 3.3.2, "prevents individual q(c|x) from drifting towards high entropy," may be too strong.) However, supplementary materials should not be necessary to grasp the logical flow of the paper. This point could be explained in more detail in the main text.
  - I apologize, but I could not fully grasp the meaning of "the codewords in our codebook are continuously updated throughout training, leading to a non-static categorical representation." at the end of p3. --> (Q2), (Q3)
  - In Figure 4, the level of entropy regularization is not clearly specified. --> (Q4)

* The experiments are not sufficient to support the claims.
  - Reconstruction accuracy is measured solely by MSE. However, for image-related tasks such as compression and generation, perceptual metrics are also important. --> (Q5)
  - Showing examples of reconstructed images would further demonstrate the effectiveness of the proposed method. --> (Q6)
  - Since the proposed methods utilize the Gumbel-Softmax technique similar to existing work, there should be, more or less, some degree of gradient estimation error. Demonstrating that the proposed method reduces this error would provide stronger support for the claims. --> (Q7)

**Questions:**

(Q1) What is the key difference between your work and those of Takida et al. (2022) and Williams et al. (2020), regarding adherence to the Bayesian framework?

(Q2) What is meant by a non-static categorical representation, and what are its benefits?

(Q3) How does the continuous updating of the codewords throughout training lead to this non-static categorical representation?

(Q4) Could you specify the level of entropy regularization shown in Figure 4?

(Q5) Could you provide evaluations using perceptual metrics, such as reconstruction FID?

(Q6) Could you show qualitative comparison with examples of reconstructed images?

(Q7) Could you demonstrate that the proposed method reduces the gradient estimation error caused by the Gumbel-softmax technique with some experimental results or mathematical guarantees?

---

### Official Review · Reviewer_byhT · 2024-11-03

**Soundness:** 2
**Presentation:** 3
**Contribution:** 2
**Rating:** 5
**Confidence:** 4

**Summary:**

The submission considers a VAE with a Gaussian mixture piror whose mean parameters can be seen as the codebook in VQ-VAEs, but with a non-degenerate variance on the generative and inference paths. The paper suggests a variational objective involving a mini-batch approximation of the aggregated posterior over the mixture/codebook index. The variational distributions of the mixture index can then be optimized using Gumbel-Softmax approximations. Experiments for CIFAR10 and CelebA show that the suggested approach yields improved MSE for the reconstruction errors compared to previous VQ-VAE models, as well as codebook usage in terms of perplexity.

**Strengths:**

Propagation of gradients for VQ-VAEs is a challenging problem and previous approximate approaches - such as straight-through estimators (STE), Gumbel-Softmax approximations or stochastic quantization - can lead to codebook collapse wherein a large fraction of the codebooks remain unused. In contrast, the paper considers a Gaussian Mixture prior model and claims to optimize an evidence lower bound to the model log-likelihood. The suggested model and variational objective is different from previous work that for example decode the quantised latent. The usage of min-batch approximation for the marginal distribution over the mixture component is also new as far as I am aware. Empirical results indeed confirm that the method leads to better reconstruction and codebook usage compared to VQ-VAE models with STE and different codebook lookup tricks.

**Weaknesses:**

I’m not convinced that the derivation in Appendix A2 for the ALBO bound is valid. In particular, why is the entropy of the variational distribution negative? If this is a Gaussian, does there have to be a constraint on the variance thereof?

Is the ALBO still a lower bound on the LLH even if the marginal (aggregated posterior) over c is approximated by a mini-batch approximation? Does the mini-batch size have to be sufficiently large?

The objective contains a KL penalty between the marginal $q(c)$ and the prior $p(c)$. This appears to be similar to an InfoVAE [1] bound that also regularises the marginal distribution of the latents (although commonly with continuous latents), but resort to the MMD to avoid the intractability of computing the KL involving the marginal/aggregated posterior when the latent space is continuous. Such objectives seem to be sensitive to the choice of weight for the different regularising terms. However, I feel that the effect of such hypeparameters (viz $\beta$ and $\gamma$) has not been studied in detail in this paper, for example experimentally.

The experimental validation appears a bit limited. In particular, it is not clear how well the generative performance is qualitatively? Also different metrics such as FID values would be useful. In particular, I would expect that the choice of $\beta$ and $\gamma$ affects the reported MSE values.

 [1] Zhao, Shengjia, Jiaming Song, and Stefano Ermon. "InfoVAE: Information Maximizing Variational Autoencoders." 2017

**Questions:**

In the simulations for Fig3, can you elaborate what is the ‘exact gradient’ from ‘passing each categorical one-hot action through the non-linear decoder’? For this experiment, what is the distribution of the inputs and the MLP weights?

For table 1, do all methods use a latent space with the same dimension and same number of mixtures/codebooks?

Minor:
In eq. (3), how is softmax defined and how is it different from $\text{softmax}_\tau$ in line 127?

---

### Official Review · Reviewer_ryUM · 2024-11-03

**Soundness:** 1
**Presentation:** 2
**Contribution:** 2
**Rating:** 1
**Confidence:** 3

**Summary:**

The present paper introduces an idea to introduce a mixture of gaussians as a prior in a VQ-VAE. Furthermore, they introduce what is called Aggregated Categorical Posterior Evidence Lower Bound (ALBO), which allows compatibility with Gumbel-Softmax gradient estimation. Finally, they present some experiments with CIFAR-10 and CelebA in which they provide MSE and Perplexity as performance measurements.

**Strengths:**

I believe that the paper is well contextualized and that a good general view of similar works is given throughout the paper.

I find studying the use of more complex priors an interesting topic in representation learning that can leverage the performance of models and obtain richer representations.

**Weaknesses:**

It results complicated to see how the works in the literature connect with each part of the present proposal. For example, what is the purpose of equation (10)? How does it connect to the next part of the paper? The different ideas of the paper look like unconnected patches and I miss a storyline that helps to understand the motivation of the paper.

One of the main novelties of the paper is claimed to be the use of mixture of Gaussians as prior. However, I miss in section 3 an explanation on how you introduce this prior. The only part that I can find somehow related is in line 335, but in this case $z$ is not sampled from a mixture of gaussians but from a gaussian which mean is a linear combination of different vectors. I would like the authors to clarify better how they use the mixture of gaussians as a prior, since this is an interesting topic. I would recommend the authors to provide a specific subsection or paragraph in Section 3 that explicitly defines the mixture of Gaussians prior and explains how it is incorporated into the model's generative process. This would help clarify the novelty and implementation of this key aspect of the paper.

One of the other novelties of this paper is the introduction of the ALBO, whose derivation is in Appendix A.2. However, I believe that this derivation is incorrect. In the penultimate line of the derivation what is inside the parenthesis is false. This is only true when $z$ is a discrete variable. This term is equal to $-H_q(z|x)$ and, since $z$ is a continuous variable, then it can be larger than 0, since this is a differential conditional entropy and not a Shannon conditional entropy (see https://en.wikipedia.org/wiki/Conditional_entropy#Conditional_differential_entropy:~:text=In%20contrast%20to%20the%20conditional%20entropy%20for%20discrete%20random%20variables%2C%20the%20conditional%20differential%20entropy%20may%20be%20negative). Please, revisit the derivation taking into account the distinction between discrete and continuous entropy. This will help ensure the mathematical rigor of the paper.

The experiments are a bit simple. Due to the big amount of applications that VQ-VAE can have, the paper would benefit from experiments in specific applications other than simply reconstruction performance. In fact, I don’t think that reconstruction performance reflects the performance of a VAE (and any of its variations) in almost any task. Vanilla Autoencoders tend to have a lower MSE than VAE (since they are optimized only for that purpose), but they are less powerful models.

I find some issues with mathematical notation. Some variables are not properly defined or there are some imprecisions and mistakes. I list some next:

•	In lines 93-95, maybe I misunderstood this sentence, but you could simply say that $\hat{z}$ is a single random variable with various dimensions.

•	In equation 1, I don’t think the notation \left[1_{i=j}\right]^C_{i=1} is a widely used notation. Some others are some more widely used notations (see for example https://stats.stackexchange.com/questions/436975/compact-notation-for-one-hot-indicator-vectors).

•	Also in equation 1: if j = arg min…

•	In equation 3, I believe that $\sigma$ is not defined.

•	In line 127, notation is incorrect. You do not calculate the softmax of $q_i$ but of $q$. Maybe, it would be easier to include the definition of Softmax (eq. 2 of CATEGORICAL REPARAMETERIZATION WITH GUMBEL-SOFTMAX)

•	In equation 4, why do you assume an uniform prior?

•	In line 185, y ou have not mentioned $\pi$ yet, so it does not make sense the clause "where $\pi$" represents the prior...". This sentence should be reformulated.

•	In line 270, the fact that $\sigma_x$ and $\sigma_z$ are fixed does not imply that "the objective function is constructed by minimizing the negative ALBO". If anything, it causes that maximizing the likelihood becomes equivalent to minimizing the euclidean distance. Thus, the paragraph should be reformulated.

•	In equation 14, how do you calculate this term in practice? I cannot find how you define p(c).

**Questions:**

•	In line 77, “a novel”.

•	In equation 5, is the only novelty here the use of $\epsilon$ ? It would be convenient to clarify what is the main contribution of this paper in that Markov Chain

•	In line 186-188: In previous paragraph you mentioned that $z$ follows a Gaussian mixture but in this sentence you say that you sample it from a Gaussian with mean $\mu_c$. Why is this happening and what value takes $c$ here?

---

### Note · Authors · 2024-12-07

I have read and agree with the venue's withdrawal policy on behalf of myself and my co-authors.